# Clinical and Biological Differences between Upper Tract Carcinoma and Bladder Urothelial Cancer, Including Implications for Clinical Practice

**DOI:** 10.3390/cancers15235558

**Published:** 2023-11-23

**Authors:** Félix Lefort, Yasmine Rhanine, Mathieu Larroquette, Charlotte Domblides, Luc Heraudet, Baptiste Sionneau, Simon Lambert, Matthieu Lasserre, Grégoire Robert, Alain Ravaud, Marine Gross-Goupil

**Affiliations:** 1Department of Medical Oncology, University Hospital of Bordeaux, 33000 Bordeaux, France; yasmine.rhanine@outlook.fr (Y.R.);; 2Faculty of Medicine, University of Bordeaux, 33000 Bordeaux, France; gregoire.robert@chu-bordeaux.fr; 3Department of Urology, University Hospital of Bordeaux, 33000 Bordeaux, France

**Keywords:** upper tract urothelial carcinoma (UTUC), invasive, metastatic, bladder carcinoma, systemic treatments

## Abstract

**Simple Summary:**

This review examines differences and similarities between upper tract urothelial carcinoma (UTUC) and bladder urothelial carcinoma (BUC) with respect to their epidemiological, clinical, pathological, and biological features and discusses the resulting therapeutic consequences. Systemic treatments for invasive and metastatic diseases are considered, and an overview of the expected developments in this field is provided.

**Abstract:**

Upper tract urothelial carcinoma (UTUC) is a rare disease included, along with the much more frequent urothelial bladder cancer (BUC), in the family of urothelial carcinomas (UCs). However, while UTUCs and BUCs share several features, their epidemiological, clinical, pathological, and biological differences must be considered to establish an optimal therapeutic strategy. This review examines the clinical differences between UTUC and BUC, as well as the main results obtained by molecular screening of the two diseases. The findings of clinical trials, performed in peri-operative and metastatic settings and assessing systemic treatments in UC, are summarised. A comparison of the data obtained for UTUC and BUC suggests improved therapeutic approaches, both in regards to routine practice and future drug development.

## 1. Introduction

Upper tract urothelial carcinoma (UTUC) is a rare cancer which is part of a much more frequent group of tumours known as urothelial carcinomas (UCs). Among the latter, bladder urothelial carcinoma (BUC) accounts for 90–95% of the cases [1]. While this grouping is based on the shared features of UTUC and BUC, the epidemiological, clinical, pathological, and biological differences between UTUC and BUC account for their description as “disparate twins” [2], impacting therapeutic strategies. Since the overwhelming majority of UCs are BUCs, studies leading to approved treatments for UTUC included very few UTUC patients. Thus, approval was granted by analogy with the guidelines proposed for BUC. Over the past few years, new molecules have been developed that have improved the prognosis of patients with BUC, but the data from the respective clinical trials should be more closely examined regarding the efficacy of these drugs for UTUC [3,4]. We begin this review with a comparison of the main characteristics of UTUC vs. BUC. We then analyse the data on recently approved molecules or emerging therapeutic targets in order to draw conclusions relevant to clinical practice and future research.

## 2. Epidemiology

The incidence of UTUC is approximately 2 per every 100,000 inhabitants/year [5], and that of BUC is about 18 per every 100,000 inhabitants/year [6]. The average age of UTUC and BUC patients at diagnosis is similar, around 73 years [7,8], but the male/female ratio differs: 2:1 for UTUC and 4:1 for BUC [7,9]. UTUC is more often diagnosed at an invasive stage than is BUC, with 56% and 25% of cases, respectively [7,10], a difference occurring due to the thinness of the ureteral wall, but also resulting from the more aggressive biology of UTUC. At the time of the initial diagnosis, the incidence of metastatic UTUC is only 12–16% [11], but ~30% of patients with localised UTUC will eventually develop metastases [10], a rate similar to that observed in BUC [12]. The risk of BUC recurrence is more frequent (22–47%) after UTUC [13,14] than is UTUC recurrence after BUC (2–6%) treatment [15]. This can be explained anatomically, as the ureteral meatus possesses an anti-reflux system that may prevent the dissemination of cancer cells from the bladder.

## 3. Risk Factors

Smoking is a major risk factor for UC. Studies of UTUC have estimated an increase in the relative risk from 2.5 to 7% [16,17,18], as also determined in BUC [19]. This risk varies according to smoking intensity and decreases after smoking cessation. Continued smoking after diagnosis is a poor prognostic factor [20]. Occupational exposure to aromatic amines, polycyclic aromatic hydrocarbons, and chlorinated solvents is also a risk factor for UTUC and BUC [21], as is chronic exposure to acrolein (an active metabolite of cyclophosphamide and ifosfamide) [22]. Chronic infections (bilharziasis) and inflammations are risk factors for both bladder and upper urinary tract cancers, but these lead, instead, to epidermoid carcinomas [21].) These common factors cause chronic aggression of the urothelium in both the upper urinary tract and the bladder, thus accounting for the development of cancer in both sites. However, other risks factors are specific for UTUC, providing evidence of its biological differences with BUC.

Aristolochic acid (AA) is the active element of the Aristolochiaceae family of herbaceous plants. Its accidental ingestion and its use in traditional pharmacopoeia are associated with the higher incidence of UTUC in the Balkans and on the Asian continent (Balkan endemic nephropathy, and Chinese herb nephropathy) [23]. Despite their better outcomes, patients with AA-associated UTUC are at higher risk of contralateral disease and BUC and thus, should be monitored closely [24]. A high incidence of UTUC (20–26.6% of all UCs) is also found on the southwest coast of Taiwan [25], where it is associated with peripheral vasculitis (“black foot disease”), related to the high concentration of arsenic in the water supplies [26].

Lynch syndrome, resulting from a constitutional mutation in one of the genes of the DNA mismatch repair system (*MLH1*, *MSH2*, *MSH6*, *PMS2*), predisposes patients to several cancers transmitted by autosomal dominant inheritance. In terms of its localisation, UTUC is the third most common (~5%) tumour on the spectrum of Lynch syndrome tumours, after colorectal and endometrial localisations [27]. A study of 115 UTUC patients screened for Lynch syndrome found a positivity rate of 5.2% [28]. The relative risk of developing a UTUC against in case of Lynch syndrome ranges from 14 to 22% vs. from 2.2 to 4.2% in the case of BUC [29]. The *MSH2* mutation is more commonly associated with the risk of UTUC [30].

## 4. Diagnosis

The diagnostic of UTUC, when not incidental, is mainly established because of haematuria (70–80% of the cases) [31]. Flank pain and systemic symptoms (deterioration of the general health status, fever) are frequently observed before UTUC diagnosis [32]. Ultrasonography is often performed as a descrambling examination to explore haematuria or flank pain. It allows for the detection of renal ureteral or bladder masses, as well as the measurement of hydronephrosis, but it shows a mild sensitivity and specificity; thus, it cannot replace computed tomography urography (CTU). In patients with metastatic disease, the diagnosis of BUC or UTUC relies on a biopsy taken at the most convenient site (primary tumour or metastasis site). In the early stages of the disease, however, the diagnosis of UTUC can be difficult due to its anatomic location, which will likely impact the therapeutic strategy.

For tumours discovered in the renal pelvis, UTUC must be distinguished from renal cell carcinomas. CTU is the reference imaging modality for the diagnostic workup of UTUC in patients with a creatinine clearance >30 mL/min. The entire urinary system is imaged through several acquisitions, obtained before and after the injection of contrast medium; a study during the excretory phase of contrast medium elimination should be included as well. Magnetic resonance urography can depict the entire urinary system, thus providing an alternative to CTU, especially if the latter is contraindicated [33].

Following the establishment of a diagnosis of UTUC, muscle invasion must be correctly assessed. Flexible ureterorenoscopy allows for the exploration of the entire upper urinary tract, as well as for direct visualisation and biopsy of the lesion. The sensitivity of biopsy in the diagnosis of UC is 89–95% [34]. Its reliability in predicting the tumour stage is low, with a high rate of underestimation (45% of Ta lesions are actually infiltrating tumours) [34]. Also, there is a rising concern that ureterorenoscopy increases the risk of intravesical recurrence [35], and a risk-stratified approach has been proposed to avoid this in high-risk cases [36]. Urinary cytology, based on cells obtained from the natural desquamation of the urothelial lining of the urinary tract, can be performed. Cytology is recommended in the diagnosis of UTUC, although it is less sensitive and less specific than when used in BUC. It should ideally be performed in situ (selective, during an endoscopic examination), before the injection of contrast medium. Cystoscopy is also recommended as part of the routine evaluation of UTUC because of the risk of synchronous and metachronous BUC, as described above.

## 5. Pathology

The WHO’s histological classification and tumour grading system for bladder and upper urinary tract cancers are identical to those for bladder cancer. Urothelial carcinoma is the most common form of the disease, representing 90–95% of upper urinary tract cancers, whereas squamous cell carcinoma is rare (5–7%), and adenocarcinoma is even rarer (~1% of UTUCs). A variant histology (micropapillary, squamous, sarcomatoid) is found in ~25% of UTUCs [37] and BUCs [38], and is a poor prognostic factor in both.

## 6. Molecular Biology

A genomics comparison of UTUC and BUC provides the most striking example of the “disparate twins” concept [39]. Sfakianos et al. used next-generation sequencing to compare the genomics of patients with localised high-grade UTUC (*n* = 83) and BUC (*n* = 102) [40]. While many common genes were altered in BUC and UTUC, the respective prevalence differed, with a higher rate of alterations in UTUC than in BUC for *FGFR3* (35.6% vs. 21.6%, *p* = 0.065), *HRAS* (13.6% vs. 1.0%, *p* = 0.001), and *CDKN2B* (15.3% vs. 3.9%, *p* = 0.016) and higher rates in BUC than in UTUC for *TP53* and *ARID1A*. The authors also identified a trend of differences between UTUC and BUC in terms of potential therapeutic targets such as *TSC1* (11.9% vs. 3.9%, *p* = 0.100) and *PIK3CA* (10.2%vs. 21.6%, *p* = 0.084). Necchi et al. obtained similar results in a cohort of patients with advanced-stage UTUC (*n* = 479) and BUC (*n* = 1984) [41]. *FGFR3* mutations were more frequent in UTUC than in BUC (21% vs. 14%, *p* = 0.002), but the rates of amplifications (0.4% vs. 0.5%), rearrangements (3.3% vs. 3.9%), and multiple *FGFR3* alterations (1.3% vs. 1.0%) were similar. Interestingly, *FGFR3*-altered tumours showed concomitant *PI3KCA/RAS* alterations in 26.2% of UTUC patients and 26.5% of BUC patients. An increase in *HRAS* mutations was also reported (6.9% for UTUC; 2.8%, for BUC), with most of the *HRAS*-altered tumours arising from UTUC of the renal pelvis rather than from other anatomic sites. Among other targetable alterations, *ERBB2* (*HER2*) amplification was less frequent in UTUC (2.7%) than in BUC (7.9%). The homologous recombination repair pathway was frequently altered in both UTUC (17%) and BUC (20%, *p* = 0.2), but the main actionable genes, such as *BRCA* 1 and 2, were altered in only 4.9% of BUC patients and 4.6% of UTUC patients.

As noted above, Lynch syndrome and micro-satellite instability (MSI)-high tumours are more likely to be found in patients with UTUC than in those with BUC. In the study of Necchi et al., patients with UTUC exhibited more frequent MSI-high tumours (3.4%) than did patients with BUC (0.8%; *p* < 0.001) [41]. Donahue et al. showed that Lynch-syndrome-associated UTUCs have a significantly higher tumour mutational burden (TMB) than do sporadic UTUCs, but the frequency of *FGFR* alterations is the same [42]. Interestingly, *FGFR3* alterations for Lynch-syndrome-associated UTUCs are mainly *R248C* mutations, suggesting the use of the latter as a biomarker for this population.

AA-associated UTUCs are linked with a higher TMB, including more frequent mutations in *TP53*, *NRAS,* and *HRAS* [24], whereas *FGFR 3* mutations are rare, even in the early stages of the disease. The specific mutational signatures found in AA-associated UTUCs could help to identify individual exposure to this carcinogen [43].

Muscle-invasive BUCs have been classified according to their molecular subtype [44]. The 2017 TCGA classification recognises five molecular subtypes: luminal-papillary, luminal-infiltrated, luminal, basal/squamous, and neuronal [45]. Since the classification was developed without the inclusion of any patients with UTUC, Robinson et al. applied it to a cohort of 37 UTUC patients and found that most of the tumours were of the luminal-papillary type (62.5% vs. 27.3% for BUC in the TCGA study) [46].

Nectin-4 belongs to a family of cellular adhesion molecules and is found to be overexpressed in various tumours and is associated with cancer progression and poor prognosis [47]. Nectin-4 is the target protein for drugs such as the antibody-drug conjugate (ADC) enfortumab vedotin, and it is expressed in the majority of BUCs. In an immunohistochemical analysis, 83% of the biopsies from 524 BUC patients stained positive for Nectin-4 [48], whereas its expression rate in UTUC is probably lower. In a study of 99 patients with UTUC, Nectin-4 positivity was detected in 66% of the tumours examined by immunohistochemistry (IHC) [49].

The target protein for the ADC sacituzumab govitecan is Trop-2, a cell surface glycoprotein that acts as a transmembrane transducer of intracellular (IC) calcium signals [50]. TROP2 stimulates proliferation and cellular growth in human cervical and bladder cancer cells and was shown by IHC to be expressed at high rates in UTUC (94/99 patients) [51]. A study in which various cancers were immunostained for Trop-2 reported moderate to strong Trop-2 expression in 88.3% of UTUCs (*n* = 62) and 92% of high-grade invasive BUCs (*n* = 735) [52].

## 7. Treatment

The standard of surgical treatment for muscle-invasive, high-risk or recurrent low-risk, localised UTUC is radical nephroureterectomy (RNU) [33]. The choice of surgical technique (open, laparoscopy, robot) does not seem to affect efficacy outcomes [53]. RNU is often accompanied by lymphadenectomy, although the lymphatic drainage areas of the upper urinary tract are not clearly defined. Lymphadenectomy in combination with RNU enables better staging, guides therapeutic management (adjuvant chemotherapy), and may improve survival by reducing the risk of recurrence for tumours ≥ pT2 [54]. Conservative, kidney-preserving treatment can be considered for patients with low-risk lesions, defined as unifocal tumours, tumours with potential complete resection, low-grade tumours, and the absence of infiltration on imaging examinations [55]. This option must be followed by close endoscopic surveillance (flexible ureteroscopy).

### 7.1. Systemic Treatment in the Peri-Operative Setting

The standard of care for the peri-operative treatment of BUC is cisplatin-based neoadjuvant chemotherapy [56]. The same chemotherapy regimen is adopted for UTUC because of the risk of renal impairment after radical surgery. The benefit of neoadjuvant chemotherapy is well-established in BUC, with improvements in disease-free survival (DFS) and overall survival (OS), as well as an absolute improvement of ~8% in 5-year survival [57]. However, the three randomised clinical trials investigating this therapeutic strategy [58,59,60] excluded patients with UTUC; therefore, no conclusions for these patients can be drawn. Neoadjuvant chemotherapy for UTUC has been assessed only in retrospective comparative or single-arm prospective studies. In 2020, a meta-analysis collected 848 patients, 349 of whom had been treated with a neoadjuvant regimen (mainly cisplatin) and 449 who had been treated with surgery alone. The results showed a relative 56% OS benefit for the neoadjuvant chemotherapy group compared with the surgery alone group [hazard ratio (HR) = 0.44; 95% confidence interval (CI); 0.32–0.59, *p* < 0.001] [61]. Among the patients treated with neoadjuvant chemotherapy, a complete or partial (<ypT2N0M0) pathological response was determined in 11% and 42%, respectively. These relatively low rates raise concerns about potential progression during neoadjuvant treatment. For BUC, in the VESPER trial, 28% and 41% of patients treated with dd-MVAC exhibited a complete or partial (<ypT2N0M0) pathological response, respectively [62]. The benefit of neoadjuvant chemotherapy in UTUC thus remains inconclusive and must be investigated on a case-by-case basis. It is also important to note that the VESPER trial, which demonstrated the superiority of the dd-MVAC regimen over the GEMCIS regimen, included only patients with primary tumours of the bladder.

Beyond the question of benefit, a majority of UTUC patients are not eligible for neoadjuvant chemotherapy because of the unreliability of preoperative staging and histopathology, as well as the difficulty in proving the invasive nature of the tumour based on the biopsy. For BUC, conclusive evidence for the benefits of adjuvant chemotherapy is lacking, since all of the relevant trials showed significant methodological bias [63]. For UTUC, the phase III trial POUT randomised, after radical surgery, patients with localised pT2-T4 or pTany N+ UTUC [64], with 261 participants allocated to either the surveillance arm or the adjuvant chemotherapy arm. Chemotherapy was administered during the 90 days following radical surgery and consisted of four 21-day cycles of cisplatin (70 mg/m^2^) and gemcitabine (GC) (1000 mg/m^2^ on days 1 and 8 of each cycle) or carboplatin (AUC 4.5 or AUC5) and gemcitabine (GP). The results showed an improved DFS (HR = 0.45, 95% CI 0.30–0.68; *p* = 0.0001). At 3 years, 71% (95% CI: 61–78) and 46% (95% CI: 36–56) of patients receiving chemotherapy and surveillance, respectively, were event-free. This benefit was consistent across the subgroups, even for the 28% of patients who received GP [65]. This finding is critical for clinical practice, since cisplatin eligibility drops from 49% to 19% in UTUC after radical treatment [66]. An update of OS data (secondary endpoint) in 2021 showed that 67% of patients in the surveillance group were alive after 3 years versus 79% in the chemotherapy group, but reduction in the relative risk of death did not reach statistical significance (HR = 0.72; 95% CI: 0.47–1.08; *p* = 0.11).

Given the anti-tumour activity of immune checkpoint inhibitors (ICIs) in metastatic BUC, their efficacy has been assessed in the adjuvant setting. The phase 3 Checkmate 274 trial randomised patients with muscle-invasive UC who had undergone radical surgery to receive nivolumab or placebo every 2 weeks for one year [67]. The primary endpoint was DFS, among the intent-to-treat population, and expression by ≥1% of tumour cells, among patients with programmed death ligand 1 (PD-L1). The results showed a benefit of DFS for both groups. Nivolumab was approved by the European Medicines Agency (EMA) for patients with muscle-invasive UC with tumour cell PD-L1 expression ≥1% who are at high risk of recurrence after undergoing radical resection. This trial included a significant proportion (21%) of patients with UTUC, thus exceeding the usual ratio of 5%. Upon subgroup analysis, UTUC patients did not seem to benefit from adjuvant nivolumab, even after extended follow-up, as reported at the ASCO GU 2023 Symposium [68].

### 7.2. Future Perspectives

The question of peri-operative treatment for UTUC is being addressed in several ongoing clinical trials. As discussed above, the main issue regarding neoadjuvant regimens in UTUC is the need for biopsy-based proof of muscle invasion. Since most high-grade UTUCs at biopsy are found to show muscle invasion, the issue of whether tumour grade, when used as a criterion for neoadjuvant treatment, could lead to survival improvements remains to be determined. The phase II NAUTICAL trial (number of clinical trial (NCT) 04574960) randomises patients with high-grade UTUC to neoadjuvant or adjuvant chemotherapy. Another phase II/III trial (NCT04628767) also uses the criterion of high tumour grade to evaluate neoadjuvant chemotherapy, with or without durvalumab, in patients with localised UTUC. The ABACUS-2 phase 2 trial will assess the effect of neoadjuvant atezolizumab for patients with rare histological subtypes of bladder cancer or with UTUC who are at high risk of relapse (NCT04624399) [69].

The abovementioned anti-Nectin-4 antibody-drug conjugate enfortumab vedotin, shown to be effective in metastatic BUC [4], is currently being tested in the peri-operative setting. A specific phase II trial for UTUC (NCT05775471) will enrol patients at high risk of recurrence to receive neoadjuvant pembrolizumab and enfortumab vedotin and adjuvant pembrolizumab.

As also noted above, *FGFR* alterations are a more frequent feature of UTUC, especially in the early stages of the disease, and constitute a therapeutic target. The phase III PROOF 302 trial (NCT04197986) [70] includes patients with BUC and UTUC with *FGFR3* alterations and a high risk of recurrence who received neoadjuvant cisplatin or who are cisplatin-ineligible. Patients have been randomised to the placebo group or to receive anti FGFR infigratinib for up to one year in the adjuvant setting.

Since *HER2* overexpression is frequently found in UTUC (36% of score 2 or 3+ on the HercepTest) [71], a phase II trial (NCT05917158) is currently assessing the efficacy and safety of a recombinant humanised anti-HER2 antibody-drug conjugate and a PD-1 monoclonal antibody for the adjuvant treatment of HER2-positive UTUCs after RNU.

The main trials are summarised in Table 1.

## 8. Systemic Treatment in the Metastatic Setting

### 8.1. Chemotherapy

For patients with advanced/metastatic disease, the standard method of care for those with BUC is a platinum-containing regimen, with a slight benefit for cisplatin over carboplatin. As the initial trials testing platinum did not include UTUC patients [72,73], platin regimens were applied in UTUC patients by complying with the BUC guidelines. Later, a retrospective analysis examined the impact of tumour location on survival outcomes in three RCTs that included UTUC patients: EORTC 30924 (M-VAC vs. high-dose M-VAC), EORTC 30986 (GC/carboplatin and methotrexate/carboplatin/vinblastine), and 30987 (GC-paclitaxel vs. GC, in patients fit for cisplatin). Among the 1039 patients, 161 (14.7%) suffered from UTUC. No difference in progression-free survival (PFS) or OS was observed [74], thus establishing the efficacy of the platinum regimen in UTUC.

In the second-line setting, mono-chemotherapy with taxanes was historically proposed for BUC patients, albeit based on retrospective studies, with few patients and deceptive results. In 2009, Bellmunt et al. published a phase III randomised trial comparing vinflunine (a vinca alkaloid) with best supportive care in the second line setting for 370 BUC patients. While the study did not find an OS benefit in the intent-to-treat population, a statistically significant benefit was identified when the 13 patients exhibiting significant protocol deviations were excluded. In that case, the median OS was 6.9 vs. 4.3 months (HR = 0.77; 95% CI 0.61–0.98) and the overall response rate (ORR) was 8.6%. Whether the study included patients with UTUC is unclear, as no data for this population are available.

In 2015, a prospective, observational study investigated the safety and efficacy of vinflunine in patients pre-treated with platinum-based chemotherapy [75]. Vinflunine was administered in the second line setting to 51 (66%) of the 77 patients. The median ORR was 23.4%, and the OS was 7.7 months. A 2017 subgroup analysis of the data from this study showed similar results for patients with UTUC (*n* = 18) and BUC (*n* = 59), with a median OS of 5.0 and 8.2 months and an ORR of 22.2% and 23.7%, respectively [76]. These results suggest the efficacy of vinflunine in UTUC, a treatment currently recommended in the second line setting, if immunotherapy is not feasible, or as a third- or subsequent-line treatment. A remaining question concerns the activity of vinflunine after immunotherapy, since it may potentiate the effect of subsequent chemotherapy [77]. A retrospective study of 105 patients who received vinflunine before (*n* = 44) or after (*n* = 61) immunotherapy showed an improved clinical benefit (51% and 25%, respectively, *p* = 0.020) and a trend toward OS improvement. This study included 23 (22%) patients with UTUC, but no conclusion could be drawn from this subgroup analyses.

### 8.2. Immunotherapy

In 2017, the KEYNOTE-045 study showed that, compared to mono chemotherapy, pembrolizumab significantly improved OS for BUC patients with disease progression after platinum-based chemotherapy (without avelumab maintenance) [78]. This trial included 75 (14%) UTUC patients. In the subgroup analyses, pembrolizumab was associated with a benefit over that of chemotherapy which appeared larger for UTUC (HR = 0.53; 95% CI: 0.28–1.01) than for BUC patients (HR = 0.77; 95% CI: 0.60–0.97). No data for the Lynch-syndrome status in UTUC patients were available to refine these results.

In 2020, the Javelin-100 trial randomised 700 patients without disease progression after first-line chemotherapy (4–6 cycles of GC or GP) to receive either maintenance avelumab or surveillance [79]. The study showed an OS benefit for avelumab maintenance (HR = 0.56; 95% CI: 0.40–0.79), which has since become the standard of care for BUC patients. In this trial, patients with UTUC were over-represented with 187 patients (27%), allowing for a comprehensive subgroup analysis [80], which showed a persistent trend (although less important) for OS benefit for the UTUC subgroup (HR = 0.63, 95% CI: 0.48–0.81, for patients with lower urinary tract tumours; HR = 0.90; 95% CI: 0.59–1.39, for patients with UTUC).

In the first-line setting, 374 cisplatin ineligible patients received pembrolizumab within the KEYNOTE-052 phase 2 trial. The ORR was 24% for the overall population, of which 19% of patients suffered from UTUC. The ORRs for UTUC and BUC were similar, at 22% and 28%, respectively. Based on these results and those from the KEYNOTE-361 trial, the US Food and Drug Administration (FDA), but not the EMA, approved pembrolizumab for patients with metastatic urothelial carcinoma (BUC or UTUC) who are not eligible for platinum-containing regimens.

The phase 2 IMvigor210 trial enrolled 119 patients with advanced UC who were ineligible for cisplatin to receive atezolizumab as a first-line therapy. The results led to FDA, but not EMA, approval of this regimen for cisplatin-ineligible patients with PD-L1-expressing UC or any patients who are platin-ineligible in the first-line setting, regardless of the tumour’s anatomic site. While the study included à significant proportion of UTUC patients (28%), no subgroup analyses were published.

### 8.3. Targeted Therapies

In case of progression after chemotherapy and immunotherapy (maintenance or second-line), the anti-Nectin-4 ADC enfortumab vedotin is the standard of care for BUC patients. The phase 3 EV-301 trial randomised 608 patients with locally advanced or metastatic UC who had previously received platinum-containing chemotherapy, but who had experienced disease progression during or following PD-1/L1 inhibitor treatment to receive enfortumab vedotin or chemotherapy [4]. A significant improvement in OS was determined for the enfortumab vedotin group (HR = 0.70; 95% CI: 0.56–0.89) [4]. This study included 205 (34%) patients with UTUC, among whom enfortumab vedotin was associated with a benefit over chemotherapy, as determined in subgroup analyses. Recently, results of the EV 302 trial were presented at the 2023 ESMO Symposium [81]. In this trial, 886 patients with previously untreated metastatic BUC or UTUC were included. They were randomized to receive either enfortumab vedotin plus pembrolizumab or standard chemotherapy. The results showed a benefit in PFS (HR = 0.45; 95% CI: 0.38–0.45) and OS (HR = 0.47; 95% CI: 0.38–0.58) for the enfortumab vedotin plus pembrolizumab combination. This trial included a significant number of patients with UTUC (234 patients 27%). Subgroup analyses showed PFS and OS benefits for both BUC and UTUC, and indicated that pembrolizumab plus enfortumab vedotin should become the new standard in this setting.

Patients with metastatic UC harbouring an *FGFR2* or *FGFR3* alteration were shown to benefit from treatment with a pan-FGFR tyrosine kinase inhibitor. In a phase 2 study, 99 patients with UC (23 with UTUC) pretreated with chemotherapy received 8 mg of erdafitinib daily [82]. The study showed an ORR (primary endpoint) of 40% (39% for UTUC and 48% for BUC), with a median PFS of 5.5 months (95% CI: 4.2–6.0) in the overall population; no other data are available for the UTUC subgroup. The THOR phase III trial assessed erdafitinib vs. docetaxel or vinflunine in patients with advanced or metastatic UC. Patients must have shown progression after one or two prior treatments, including therapies with an anti-PD-(L)1 agent, and tumours must had pre-specified *FGFR* alterations [83]. Erdafitinib significantly increased the median OS compared with that of docetaxel or vinflunine (12.1 months vs. 7.8 months; HR = 0.64; 95% CI: 0.47–0.88). The study population included a high proportion of UTUC patients, as 89 out of 266 (33%) patients possessed a primary tumour in the upper urinary tract. An OS benefit achieved with erdafitinib was consistently observed across the subgroups, with a greater benefit in UTUC (HR = 0.34; 95% CI: 0.18–0.64) than in BUC (HR = 0.82; 95% CI: 0.56–1.18). Erdafitinib is currently approved by the EMA for patients with advanced or metastatic UC, characterised by *FGFR* alterations, that has progressed despite chemotherapy and immunotherapy, regardless of the primary site. Given the higher incidence of *FGFR* alterations in UTUC and the clinical activity observed in this population, erdafitinib can be considered as the treatment of choice for UTUC.

### 8.4. Future Perspectives

Clinical trials dedicated to metastatic UTUC are very rare, but several molecules are currently being studied in trials that include both BUC and UTUC patients. These trials are summarised in Table 2 and Table 3.

#### 8.4.1. Trop-2

In the phase 2 mono-arm TROPHY-U-01, 113 patients with metastatic UC and disease progression after prior platinum-based and anti PD(L)-1 therapies were allocated to receive sacituzumab govitecan, an anti-Trop2 antibody conjugated to SN-38 (an active metabolite of irinotecan) [84]. While the inclusion criteria allowed for the admission of patients with UTUC, no data for this population have been published. The phase 3 TROPiCS-04 is currently assessing the efficacy and safety of sacituzumab-govitecan in patients with metastatic UC and disease progression after prior platinum-based and anti PD(L)-1 therapies (NCT04527991) [85]. The study allowed for the admission of patients with UTUC and should provide results for this subgroup.

#### 8.4.2. Immunotherapy

The results of the development of immunotherapy for UTUC and BC are currently indissociable, as there is no specific trial for UTUC. Ongoing trials with immunotherapy are evaluating several combinations of ICIs, or ICIs with other molecules, in the first-line setting as maintenance, or in the late stages of the disease (Table 2). The molecular differences between BUC and UTUC may one day allow for predictions of the ICI response and the development of biomarker-based clinical trials.

#### 8.4.3. MSI-High Tumours

Contrary to the subgroup analyses of the neoadjuvant trial Checkmate 274, the outcomes were better for UTUC than for BUC in the KEYNOTE-045 trial. These differences reflected the presence among the UTUC population of patients with MSI-high tumours, known to be very good responders to ICIs [86]. To date, there is no large dataset for ICI efficacy in patients with MSI-high metastatic UTUC, but a report on a population of ten such patients treated with ICIs showed an impressive ORR of 90%, with 100% of the patients presenting without disease progression at 15 months [87]. In the future, such patients should be screened in a clinical trial to more fully understand the subgroup outcomes.

Some trials for UC in general are also of specific interest for UTUC because of its unique biology, as noted in previous sections. This issue is further examined below.

#### 8.4.4. FGFR

The promising clinical activity of erdafitinib in UCs with *FGFR* alterations is particularly interesting for patients with UTUC, as *FGFR* alterations are more frequent in these tumours. New anti FGFR inhibitors, such as ICP-192 (gunagratinib) or TYRA-300, are currently being evaluated for UC in phase 2 trials (NCT04492293 and NCT05544552). Other anti-FGFR agents, such as AZD4547 in combination with tislelizumab (anti PD-1) and futibatinib in combination with pembrolizumab, are being tested in association with ICIs to enhance the anti-tumour effect in UC. Both are currently being evaluated in phase 2 trials (NCT05775874 and NCT04601857).

#### 8.4.5. HER2

If HER2 amplifications are of low frequency in UC and even lower in UTUC, then the development of new antibody-drug conjugates targeting low-HER2 tumours may offer new treatment opportunities for UC. A recent study reported that 64% of 130 UTUC tumours analysed by IHC were at least HER2 1+ [88]. MRG002 (trastuzumab-vedotin) an antibody-drug conjugate targeting HER2 is being tested in the second- or third-line setting in a randomised phase 3 trial (NCT05754853) for patients with metastatic UC with HER2 positivity (IHC 3+ or IHC 2+).

#### 8.4.6. The Homologous Recombination Repair (HRR) Pathway

The HRR pathway is frequently altered in both BUC and UTUC, suggesting the efficacy of poly(ADP-ribose) polymerase (PARP) inhibitors in these patients. In the mono-arm phase II TALASUR trial (NCT04678362), talazoparib was added to avelumab (regardless of HRR mutations) as a maintenance treatment in patients with metastatic UC without disease progression after chemotherapy consisting of a first-line platinum-regimen [89]. To improve patient selection, another mono-arm phase 2 trial selected patients with UC harbouring DNA damage response gene alterations and with disease progression, despite at least one prior line of treatment (NCT03448718). The results of these trials are likely to be very interesting for patients with AA-associated UTUC, which is often associated with HRR deficiency [90].

#### 8.4.7. HRAS

*HRAS* mutations, although rare, are twice as frequent in UTUC than in BUC. Tipifarnib is a quinolinone that inhibits the enzyme farnesyl protein transferase and prevents the activation of *Ras* oncogenes. A phase 2 mono-arm trial is currently assessing tipifarnib in UCs harbouring *HRAS* or *STK11* mutations for patients pre-treated with platinum-based chemotherapy (NCT02535650). In preliminary results from 21 patients, the ORR was 24%, but there was no response for patients with tumours harbouring STK11 mutations [91].

#### 8.4.8. TSC1

*TSC1* mutations are three times more frequent in UTUC than in BC. Sapanisertib is a dual mTORC1/2 inhibitor that was tested in a phase 2 mono-arm trial (NCT03047213) in patients with metastatic UC. However, due to the absence of an objective response and poor tolerance of the drug, the trial was suspended [92].

**Table 2 cancers-15-05558-t002:** The main phase 2 trials for metastatic UTUC currently enrolling or for which results are pending.

Study Name and/or Number	Population	Experimental Arm	Comparator Arm	Primary Endpoint	Current Status
NCT05219435	- Stable after 4–6 cycles of first-line platinum based therapy	Nivolumab + ipilimumab	NA	PFS	Recruiting
NCT04678362 [89]	- Stable after 4–6 cycles of first-line platinum based therapy	Talazoparib + avelumab	NA	PFS	Recruiting
NCT03448718	- Progression despite one prior line of treatment for metastatic UC - Somatic alteration considered pathogenic/likely pathogenic in a predetermined list of DDR genes	Olaparib	NA	ORR	Active; not recruiting
NCT05775874	- Unresectable locally advanced or metastatic UC - *FGFR2/3* alterations	AZD4547 (Anti FGFR) + tislelizumab (Anti PD1)	NA	Safety index/ORR	Recruiting
NCT04601857 [93]	- First-line setting - Unfit for standard platinum-based chemotherapy. - Cohort A: *FGFR3* mutation or *FGFR1-4* fusion/rearrangement - Cohort B: all other patients with UC	Futibatinib (anti FGFR) + pembrolizumab	NA	ORR	Recruiting
BAYOU NCT03459846	- First-line setting - Ineligible for platinum-based chemotherapy - Known tumour HRR mutation	Arm 1: durvalumab/placebo Arm 2: durvalumab/olaparib	NA	PFS	Active; not recruiting
NCT02122172	- Prior platinum-based chemotherapy regimen - Second-line setting - Regardless of EGFR or HER2 expression	Afatinib	NA	PFS	Recruiting
NCT03047213 [92]	- Prior platinum-based chemotherapy regimen or cisplatin unfit - Tumours harbouring a *TSC1* or *TSC2* mutation	Sapanisertib	NA	ORR (tsc1 patients)	Active; not recruiting
PRESERVE3 NCT04887831	- First line setting	Trilaciclib + gemcitabine + cisplatin or carboplatin followed by trilaciclib i avelumab maintenance	Gemcitabine + cisplatin or carboplatin followed by avelumab maintenance	PFS	Active; not recruiting

DDR: DNA damage response and repair; HRR: homologous recombination repair; NA: not applicable; ORR: overall response rate; PFS: progression-free survival; UC: urothelial carcinoma.

**Table 3 cancers-15-05558-t003:** The main phase 3 trials for metastatic UTUC currently enrolling or for which results are pending.

Study Name and/or Number	Population	Experimental Arm	Comparator Arm	Primary Endpoint	Current Status
NCT05911295	- Unresectable locally advanced or metastatic UC - First line setting - Patients platin-eligible - HER2 expression ≥ 1+ by immunohistochemistry	Disitamab vedotin + pembrolizumab	Gemcitabine + cisplatin or carboplatin	PFS	Recruiting
NCT05754853	- Progression following a platinum-containing regimen and (PD-1/PD-L1) therapy - HER2-positive (IHC 3+ or IHC 2+)	MRG002 (trastuzumab vedotin)	Physician’s choice of treatment (docetaxel/paclitaxel/gemcitabine hydrochloride/pemetrexed disodium)		Recruiting
EV302 NCT04223856	- First-line setting	Arm A: enfortumab vedotin + pembrolizumab Arm C: enfortumab vedotin + pembrolizumab + cisplatin or carboplatin	Gemcitabine + cisplatin or carboplatin	PFS	Active; not recruiting
TROPICS-04 NCT04527991	- Progression following a platinum-containing regimen and (PD-1/PD-L1) therapy	Sacituzumab govitecan	Physician’s choice of treatment (taxol/taxotere/vinflunin)	OS	Active; not recruiting
THOR trial NCT03390504	Cohort 1: - Prior treatment with anti-PD-(L)1 - No more than two prior lines of systemic treatment Cohort 2: - No prior treatment with an anti-PD-(L)1 agent - Only one line of prior systemic treatment	Erdafitinib	Vinflunine or docetaxel	OS	Active; not recruiting
NCT03898180	- Cisplatin-ineligible with a PD-L1-CPS ≥ 10 - Ineligible for any platinum-containing chemotherapy, regardless of CPS - First-line setting	Arm A: pembrolizumab + lenvatinib Arm B: pembrolizumab monotherapy	Pembrolizumab + placebo	PFS	Active; not recruiting

NA: not applicable; OS: overall survival; ORR: overall response rate; PFS: progression-free survival; PD(L)1: programmed cell death protein 1 (ligand); UC: urothelial carcinoma.

## 9. Discussion

While the common features of BUC and UTUC suggest shared therapeutic targets, the differences between these tumours should be taken into account in clinical practice and in trial design.

In the neoadjuvant setting, it is tempting to extrapolate the benefit of a neo adjuvant cisplatin-based regimen demonstrated in BUC to UTUC, especially because many patients will become cisplatin ineligible after nephroureterectomy. However, several issues specific to UTUC merit consideration. First, unlike BUC, there is no level 1 evidence for the benefit of neoadjuvant chemotherapy in UTUC. In 2022, a systematic review of 24 studies using neoadjuvant therapy in UTUC were analysed. Neoadjuvant treatment seemed to be associated with improved survival and better pathological response compared to the results for surgery alone. However, this result applied to retrospective or single arm trials, and there was no clear advantage when this method was compared to surgery followed by adjuvant treatment [94]. The lower ORR observed in UTUC when compared to those in BUC (determined in retrospective studies) raises concerns regarding the risk of tumour progression during neoadjuvant treatment and makes the side effects less acceptable. The use of biomarkers to predict the response to neoadjuvant treatment will improve patient selection. An analysis of the ORR for cisplatin-based chemotherapy, according to various molecular signatures (DNA repair genes, molecular subtypes, regulators of apoptosis, or genes involved in cellular efflux), failed to show that any were strong enough to be used in clinical practice [95]. The results of ongoing neoadjuvant trials should help to refine the indications for neoadjuvant therapy, especially for tumours harbouring targetable molecules.

The second main issue for neoadjuvant treatment in UTUC is the need to clearly identify muscle invasion, since the biopsies are much narrower and more difficult to perform than in BUC. A correlation with tumour grade was reported, as muscle invasive tumours at nephroureterectomy were found in 60% of patients with biopsies showing high-grade tumours [96]. Thus, several ongoing neoadjuvant trials proposed high-grade as an inclusion criterion. Nomograms using clinical biological and pathologic features, with an accuracy in predicting muscle-invasive disease of ~80% [95,96], are available and could be useful tools for identifying candidates for clinical trials. Other predictive factors based on imaging and molecular biology studies mays also eventually help to predict muscle invasion more effectively.

In the adjuvant setting, the benefit of platin-based chemotherapy was well demonstrated in the POUT trial. The DFS benefit was significant for patients who received cisplatin or carboplatin, a crucial finding for clinical practice, since most patients exhibit renal impairment after nephroureterectomy. The Checkmate 274 trial showed that nivolumab improved DFS for the overall population in the adjuvant setting, but subgroup analyses showed no benefit for UTUC patients. Since most UTUCs are of the luminal-papillary molecular subtype, characterised by immune cell infiltration, they are probably less responsive to immunotherapy [45]. Further investigation is needed to determine the precise role of adjuvant immunotherapy for UTUC patients, especially because this indication competes with that used for adjuvant chemotherapy (as concluded in the POUT trial). A meta-analysis suggested a greater benefit of chemotherapy over immunotherapy in this setting [96]. Also, patients with UTUC associated with Lynch syndrome are more likely to benefit from immunotherapy, in which case, it may be more important to consider the MSI status than the primary site.

In the metastatic setting, the anatomic specificities of UTUC are a less informative determinant of the therapeutic strategy, and clinical trials have often mixed UTUC and BUC patients. However, the biological differences between the two entities, as discussed herein, can be useful in clinical practice. For instance, a higher proportion of UTUCs than BUCs are MSI-high tumours. The MSI-high status should then be assessed for UTUC, since it can predict immunotherapy efficacy, but also the screening of patients and their families for germline mutations should also be recommended. While several targetable gene alterations are over-represented in UTUC compared to BUC, they are nonetheless generally present in both diseases. Thus, the rarity of dedicated trials for metastatic UTUC is not an issue, if these patients can be included in trials gathering all UCs. Nevertheless, since the UTUC population is likely to exhibit distinct responses in clinical trials, the respective subgroup data and analyses should be systematically presented.

## 10. Conclusions

Although the similarities between UTUC and BUC have allowed for the rapid development and use of effective therapies in this rare group of diseases, the more recent understanding of the nature of these “disparate twins” raises critical issues concerning UTUC treatment. The lack of substantial evidence for neoadjuvant chemotherapy in UTUC has to be taken into account in routine practice, and there is an unmet need for dedicated trials in this setting. Comprehensive data from UTUC subgroup patients in mixed clinical trials should also be systematically published. Therapeutic strategies using molecular targets specific to UTUC could also lead to more precise medicine and improved outcomes for these patients.

## Figures and Tables

**Table 1 cancers-15-05558-t001:** The main phase 3 trials for perioperative UTUC currently enrolling or for which results are pending.

Study Name and/or Number	Phase	Population	Experimental Arm	Comparator Arm	Primary Endpoint	Current Status
URANUS NCT02969083	Phase 2 Randomised Neoadjuvant Adjuvant	- cT2-pT4 cN0-N1 M0 - Randomisation between ARM A and B for eligible patients - RNU for ineligible patients	ARM A: RNU ARM B: neoadjuvant chemotherapy ARM C: adjuvant chemotherapy	NA	% of patients randomised	Recruiting
PROOF 302 NCT04197986 [70]	Phase 3 Randomised Adjuvant	- Invasive localised UTUC with FGFR3 alteration - If neoadjuvant chemotherapy, Stage ≥ ypT2 and/or yN+	Infigratinib	Placebo	DFS	Not recruiting
NCT05917158	Phase 2 Adjuvant	- pT2-pT4 pN0-3 M0 or pTany N1-3 M0 - Tissue immunohistochemistry HER2 2~3+	RC48-ADC (Anti Her2 ADC) + JS001 (anti-PD1)	NA	DFS	Recruiting
NAUTICAL NCT04574960	Phase 3 Randomised Neoadjuvant	- cT1-4 N0 M0 and high grade	Neoadjuvant chemotherapy	Adjuvant chemotherapy	DFS	Recruiting
NCT05775471	Phase 2 Neoadjuvant Adjuvant	- High-risk localised UTUC	Pembrolizumab + enfortumab Vedotin (néoadjuvant) followed by pembrolizumab (adjuvant)	NA	ORR	Not yet recruiting

ADC: antibody-drug conjugate; DFS: disease free survival; NA: not applicable; ORR: overall response rate; PD1: programmed cell death protein 1; RNU: radical nephroureterctomy; UC: urothelial carcinoma.

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
