# Peer review of "Clinical and Biological Differences between Upper Tract Carcinoma and Bladder Urothelial Cancer, Including Implications for Clinical Practice"

_cancers, 2023, doi:10.3390/cancers15235558_

Round 1

Reviewer 1 Report

Comments and Suggestions for Authors

1.     The surgery strategy for UTUC and BUC is quit different. The systemic treatment pre-/post- operative according to different stage of tumor is vary. These differences could be discuss in this review.

2.     As mentioned in page 15, phase 2 clinical test showed Antibody-drug conjugate (ADC) (anti-Trop-2 antibody) strategy. Some clinical trials shown ADC could provide a promising efficacy with a manageable safety profile in patients with HER2+locally advanced or mUC who had failed at least one line of systemic chemotherapy. How author expect the application of ADC in the future management of BUC/UTUC? Also for those patients diagnosed with T2+ BUC.

3.     Please make the format consistently.

4.     It would be more logically to have a brief introduction contained before the knowledge of specific genes (Page 15 between future perspectives and Trop-2 paragragh). 

Author Response

The surgery strategy for UTUC and BUC is quite different. The systemic treatment pre-/post- operative according to different stage of tumor is vary. These differences could be discuss in this review.

I agree but we focused the discussion on systemic treatments because this review is part of a special issues in which surgical treatment of UTUC should be discussed

Thus the part about surgery is more to add context in our review to discuss  perioperative systemic treatments

2.     As mentioned in page 15, phase 2 clinical test showed Antibody-drug conjugate (ADC) (anti-Trop-2 antibody) strategy. Some clinical trials shown ADC could provide a promising efficacy with a manageable safety profile in patients with HER2+locally advanced or mUC who had failed at least one line of systemic chemotherapy. How author expect the application of ADC in the future management of BUC/UTUC? Also for those patients diagnosed with T2+ BUC.

We discuss ADC as a future perspective in the peri operative setting, line 284 for HER2, and line273 for EV

And in the metastatic setting (line 410 for trop 2 and line 450 for  HER2)

We think ADC will develop widely in the future, in combinations and in earlier setting.

To complete the review, we add the data of EV302 phase III trial presented at ESMO 2023 congress, with interesting results for UTUC in subgroup analyses

3.   Please make the format consistently.

If you refer to tables they will be rearranged by the editor

We also pointed problems with fonts that are different for references (e.g. [1]) and fixed it

4. It would be more logically to have a brief introduction contained before the knowledge of specific genes (Page 15 between future perspectives and Trop-2 paragragh). 

They are detailed before in the “molecular biology” section. But to be clearer we added sentences to detail  they roles in cell biology

Reviewer 2 Report

Comments and Suggestions for Authors

This article discusses upper tract urothelial carcinoma (UTUC), a rare type of cancer that belongs to the group of urothelial carcinomas (UCs). The article covers various aspects related to UTUC, such as epidemiology, risk factors, diagnosis, pathology, molecular biology, treatment, and future perspectives.

Here are some aspects of the article that need improvement:

  • Clarity: The article contains several long sentences and complex phrases, making it difficult to understand. It would be helpful to break down the information into smaller, more manageable sentences for improved readability. Some sentences are incomplete, and they need to be revised to make sense. For example, "Therefore, no conclusions for these patients can be drawn from the data." The article should aim for greater clarity and conciseness in its language. Complex phrases and wordiness can be simplified for better readability.
  • Abbreviations: The article uses several abbreviations (e.g., RNU, OS, DFS) without explaining them. It's important to introduce and define these abbreviations when first mentioned to aid reader comprehension.  The article should maintain consistency in the use of terminology and terms. For example, it alternates between "UTUC" and "upper tract urothelial carcinoma." It's important to use consistent terminology throughout the article.
  •                   Language: The article could benefit from language editing to improve grammar and phrasing. Some sentences are awkwardly constructed, making it challenging to grasp the intended meaning.
  • Discussion: The lack of substantial evidence for neoadjuvant chemotherapy in UTUC should be emphasized as a critical issue. It's crucial to clearly state the limitations of current knowledge and the need for further research to establish its efficacy. This disparity in ORR between UTUC and BUC should be highlighted as a significant concern, and the potential reasons for this difference should be explored. Moreover, It's crucial to stress that due to these biological differences, presenting separate subgroup data for UTUC patients in clinical trials is essential for accurate assessment and treatment strategies. he article mentions ongoing clinical trials and future perspectives, but it doesn't provide details or sources for these trials and lacks proper citations and references to validate the information presented. Adding information about these trials and references to their sources would enhance the credibility of the article. Please add and discuss also those interesting paper on the topic: PMID: 35383431; PMID: 37680219; PMID: 35412680.
  • Conclusion: The conclusion should recap the critical issues and the need for further research or improvements in the treatment of UTUC. Please expand it.

Overall, the article needs significant improvements in terms of structure, clarity, and proper citation of sources to enhance its overall quality.

Author Response

Remarks

Author’s answers

Clarity: The article contains several long sentences and complex phrases, making it difficult to understand. It would be helpful to break down the information into smaller, more manageable sentences for improved readability. Some sentences are incomplete, and they need to be revised to make sense. For example, "Therefore, no conclusions for these patients can be drawn from the data." The article should aim for greater clarity and conciseness in its language. Complex phrases and wordiness can be simplified for better readability.

Thank you for your remarks.

We reworked the article to shorten sentences.

Plus, as detailed below, the article was corrected by medical writers

·         Abbreviations: The article uses several abbreviations (e.g., RNU, OS, DFS) without explaining them. It's important to introduce and define these abbreviations when first mentioned to aid reader comprehension.  The article should maintain consistency in the use of terminology and terms. For example, it alternates between "UTUC" and "upper tract urothelial carcinoma." It's important to use consistent terminology throughout the article.

Thank you for your remark:

Abbreviations were detailed:

OS line 208

RNU line 192

DFS line 207

And UTUC always used after first occurrence (line 30) except for line 69 because we refer to anatomic site (bladder and upper tract) and not to the disease (urothelial carincoma)

   Language: The article could benefit from language editing to improve grammar and phrasing. Some sentences are awkwardly constructed, making it challenging to grasp the intended meaning.

This article has been reviewed by medical writers (textcheck) who are native English speakers.

 They corrected spelling, grammar but also style, to shorten sentences.

·         Discussion: The lack of substantial evidence for neoadjuvant chemotherapy in UTUC should be emphasized as a critical issue. It's crucial to clearly state the limitations of current knowledge and the need for further research to establish its efficacy.

·          

·         This disparity in ORR between UTUC and BUC should be highlighted as a significant concern, and the potential reasons for this difference should be explored.

·          

·         Moreover, It's crucial to stress that due to these biological differences, presenting separate subgroup data for UTUC patients in clinical trials is essential for accurate assessment and treatment strategies.

·          

·         The article mentions ongoing clinical trials and future perspectives, but it doesn't provide details or sources for these trials and lacks proper citations and references to validate the information presented. Adding information about these trials and references to their sources would enhance the credibility of the article.

·          

·         Please add and discuss also those interesting paper on the topic: PMID: 35383431; PMID: 37680219; PMID: 35412680.

·          

Thank you for these useful remarks.

The disparity in ORR is detailed in “Systemic treatment in the peri-operative setting” section (lines 217-219) and in discussion section (line 501-503)

We agree with you that the importance to presenting separate subgroup data for UTUC patients in clinical trials is essential for accurate assessment and treatment strategies. Thus it is detailed in the discussion section (line 548-550) and we also added a statement to emphasize it in the conclusion.

We referenced ongoing trials with NCTs. Based on your remarks, we added references for ongoing trials who presented abstract about design or partial results.

Thank you for your interesting propositions:

PMID: 35383431; I added a reference to this paper in the discussion section.

PMID: 35412680 and PMID: 37680219:

I cited these interesting reviews in the “diagnostic” and “Treatment” sections but without detailing them too much because this review is part of a special issue on UTUC, and surgical procedures will be detailed in another review.

·         Conclusion: The conclusion should recap the critical issues and the need for further research or improvements in the treatment of UTUC. Please expand it.

·          

Overall, the article needs significant improvements in terms of structure, clarity, and proper citation of sources to enhance its overall quality.

·          

We expended the conclusion to emphasize the

critical points you discussed previously

Round 2

Reviewer 2 Report

Comments and Suggestions for Authors

The manuscript has been significantly improved compared to previous versions and is now an engaging and informative read. The topics discussed are extremely interesting, and the author has presented a unique and well-documented perspective.